# Spread of *Jacobaea vulgaris* and Occurrence of Pyrrolizidine Alkaloids in Regionally Produced Honeys from Northern Germany: Inter- and Intra-Site Variations and Risk Assessment for Special Consumer Groups

**DOI:** 10.3390/toxins12070441

**Published:** 2020-07-07

**Authors:** Christoph Gottschalk, Florian Kaltner, Matthias Zimmermann, Rainer Korten, Oliver Morris, Karin Schwaiger, Manfred Gareis

**Affiliations:** 1Chair of Food Safety, Faculty of Veterinary Medicine, Ludwig-Maximilian-University Munich, Schoenleutnerstr. 8, 85764 Oberschleissheim, Germany; florian.kaltner@lmu.de (F.K.); matths.zimmermann@web.de (M.Z.); karin.schwaiger@lmu.de (K.S.); manfred.gareis@lmu.de (M.G.); 2Chair of Analytical Food Chemistry, Technical University of Munich, Maximus-von-Imhof-Forum 2, 85354 Freising, Germany; 3Interessengemeinschaft Jakobskreuzkraut, Zarnekauer Siedlung 2, 23701 Zarnekau, Germany; r.korten@zvo.com (R.K.); imkereimorris@aol.com (O.M.)

**Keywords:** honey, pyrrolizidine alkaloids, *Jacobaea vulgaris*, *Senecio*, LC–MS/MS, food safety

## Abstract

Pyrrolizidine alkaloids (PA) and PA *N*-oxides (PANO) are secondary plant metabolites exhibiting genotoxic and carcinogenic properties. Apart from the roots and leaves, PA/PANO are particularly present in pollen and nectar. Therefore, the spread of *Jacobaea vulgaris* in certain regions of northern Germany has an impact on the safety of honey produced in that region. In this study, raw honey samples (*n* = 437) were collected from usually three individual beehives per site (*n* = 73) in the district of Ostholstein and analyzed for 25 PA/PANO. The results reveal mean levels of 8.4, 1.5, and 72.6 µg/kg and maximum levels of 111, 59.4, and 3313 µg/kg, depending on the season (summer 2015 and spring/summer 2016, respectively). As far as individual data are concerned, sites near areas with *J. vulgaris* growth did not necessarily result in high PA/PANO values. Furthermore, intra-site investigations revealed remarkable differences in PA/PANO levels of raw honey collected by different bee colonies at the same site. Consumption of these regionally produced honeys entails an increased exposure to PA/PANO, especially in children and high consumers. Margin of exposure values of <10,000 and an exceedance of the health-based guidance value highlight that regionally produced and marketed honey must be considered with care for a proper risk assessment and risk management.

## 1. Introduction

Pyrrolizidine alkaloids (PA) and their *N*-oxides (PANO) are secondary plant metabolites that are primarily synthesized to protect the plants from herbivorous insects. More than 600 different compounds are considered to occur in approximately 3% of all flowering plants worldwide, particularly in the species of the Asteraceae, Boraginaceae and Leguminosae. The highest contents of PA/PANO are regularly found in the seeds and flowering tops of the plants [1,2,3]. 1,2-unsatured PA/PANO with a common core structure (1-hydroxymethyl-1,2-didehydropyrrolizidine, necine base), mostly of the retronecine-, heliotridine- and otonecine-type, possess genotoxic and carcinogenic properties in humans and animals [4,5]. The broad variety of structures is reflected by different esterification at the C1 and C7-positions of the pyrrolizidine core structure with mono- or dicarboxylic acids (necic acids), forming monoesters, diesters or cyclic diesters. Their toxicity is regarded to depend on the type of esterification with cyclic diesters possessing the highest potency [2,6]. Hepatic cytochrome-P450 monooxygenases induce the transformation of ingested PA/PANO to highly reactive pyrrolic esters that rapidly bind to the cellular nucleophiles of proteins and the DNA, resulting in protein adducts, DNA–protein adducts and DNA crosslinks [7,8]. An acute intoxication regularly provokes a pathognomic hepatic sinusoidal obstruction syndrome (HSOS, or veno-occlusive disease, VOD) while long-term exposure causes liver cirrhosis, cancer and adverse effects on the lung (pulmonary arterial hypertension, PAH) and other organs [9]. Therefore, human and animal exposure to PA/PANO should generally be kept as low as possible [10,11]. 

In the light of food safety, (herbal) teas, culinary herbs and spices as well as food supplements can be contaminated with PA/PANO via co-harvesting parts or seeds of PA-containing plants and via horizontal transfer in the soil [12]. These food commodities are currently regarded as the main sources of human exposure to PA/PANO [4,13,14]. Food of animal origin, such as milk and meat, may also contribute to the consumers’ exposure if animals are fed with PA/PANO-containing feed [4,15]. Furthermore, it is known for decades that honeybees collecting the pollen and nectar of such species transfer the toxic compounds into honey. Deinzer et al. (1977) and Culvenor et al. (1981) first reported on PA/PANO-contaminated honey originating from *Jacobaea vulgaris* (syn. *Senecio jacobaea*, tansy ragwort) in the U.K. and from *Echium plantagineum* growing in Australia, respectively [16,17]. Since then, several studies worldwide (summarized in chronological order in Appendix A) focused on developing methods and identifying PA/PANO in honey from known plant sources and geographical origins (examination of raw/bulk or regionally produced honey) [18,19,20,21,22,23,24,25,26,27,28,29] and in honey from retail [19,20,21,24,28,30,31,32,33,34,35,36,37,38]. Raw and regionally produced honeys showed higher PA/PANO levels than blended retail honeys, while the latter usually showed higher rates of contamination. South European and South American honeys regularly contained more PA/PANO than middle or north European honeys, showing typical patterns of PA/PANO from predominately *Echium*, *Heliotropium*, or *Senecio* spp., depending on the geographical origin [20]. 

In 2011, a broad set of data from analyses launched by the German Honey Association was evaluated by the German Federal Institute for Risk Assessment (BfR) [39]. Health risks were assessed by using the margin of exposure (MOE) approach which is regarded as the most suited procedure in terms of a long-term exposure to genotoxic compounds such as PA/PANO and to derive the potential need of management actions [11,40]. Hereby, an MOE of >10,000 would be regarded to be of little concern for an additional cancer risk of the consumer [4,41]. The health risk assessment of the BfR was based on a benchmark dose lower confidence limit of 10% (BMDL_10_) of 73 µg/kg of body weight (b.w.), which was derived from a study with rats chronically fed lasiocarpine [42]. The results revealed an elevated risk of adverse health effects for children and high consumers of honey (MOE < 10,000) [39]. This was in line with the results of the scientific opinion of the European Food Safety Agency (EFSA) on the occurrence of PA/PANO in food and feed [11].

To date, the MOE is calculated on the basis of a BMDL_10_ of 237 µg/kg b.w./day originating from a chronic exposure experiment with riddelliine in rats [43]. Hereby, as a pragmatic approach, both EFSA and BfR ratified the sum of all detected PA/PANO as suited for calculating the PA/PANO exposure, regardless of a different toxicity of singular compounds or recently discussed relative potency factors [44]. The PA/PANO sum content is also the basis for the assessment of an acute exposure in the light of non-neoplastic health effects (hepatocyte cytomegaly). In 2008, the British Committee on Toxicity of Chemicals in Food (COT) proposed a health-based guidance value (HBGV) of 0.1 µg/kg b.w./day, which has been derived from a no-observed-adverse-effect level (NOAEL) of 10 µg/kg b.w./day of a riddelliine study with rats [43], applying a safety factor of 100 [45]. This HBGV has been adopted by the BfR [41]. 

Following the findings on the higher contamination levels in raw honeys, the BfR highlighted in its opinion a need of data on the PA/PANO-contamination of unblended, regionally produced and marketed honey as well as on geographical and botanical influences [39]. Unblended local honey can represent a higher risk of PA/PANO-exposure for consumers of such products if PA/PANO-containing plants grow within the area of residence of the beekeepers. In Middle Europe, indigenous *Senecio* species of the Asteraceae family such as *S. vulgaris* (common groundsel), *S. aquaticus* (marsh ragwort), the invasive neophyte *S. inaequidens* (narrow-leaved ragwort), and *J. vulgaris* represent important PA/PANO sources [46,47], along with *Eupatorium* spp. and certain species of the Boraginaceae family [26]. Certain regions of Germany are subject to an immense spread of *Senecio* plants during the last years. In east Schleswig-Holstein, *J. vulgaris* turned out to be a severe problem for beekeepers as well as a potential threat for the health of grazing animals (Figure 1). In 2014, the official food surveillance of the district of Eutin, Ostholstein, revealed PA/PANO levels of up to 2765 µg/kg in local honey [48].

This research project was a collaboration work with local beekeepers from the area of Eutin who provided their honey for analysis in a two-year consecutive study. The study aimed at investigating PA/PANO levels in unblended raw honey and in evaluating variations between years/seasons, sites as well as between singular beehives at the same site. Therefore, samples were directly taken from individual beehives and the geographical data and distances to areas with an immense *J. vulgaris* growth were monitored from altogether 73 locations. The data obtained were used to assess acute and chronic health risks for consumers of regionally produced and marketed honeys.

## 2. Results

### 2.1. Performance of the Method

A sensitive LC–MS/MS method based on a previously published procedure [24] was established and in-house validated for 25 PA/PANO in honey. Recoveries were determined in triplicate at three levels of fortification. The mean recoveries ranged between 82 and 121% for europine *N*-oxide and retrorsine *N*-oxide, respectively. Method’s precision expressed as relative standard deviations (intraday repeatability, RSDr) ranged between 2.9 to 22.6% for erucifoline and seneciphylline *N*-oxide. The analytical limits were calculated from between 0.01 and 0.19 µg/kg (LOD) and 0.03 and 0.59 µg/kg (LOQ), depending on the analyte (Appendix A). The linearity of the calibration curve was given from the LOD to 50 ng/mL. The method was therefore regarded to comply with the prerequisites of Commission Regulation (EC) No 401/2006 laying down methods and performance criteria for contaminants in food.

### 2.2. Occurrence of PA/PANO

Across all samples (*n* = 437), 88% contained at least one PA/PANO compound. With respect to the three different seasons, the rate of contamination was quite comparable; however, the measured PA/PANO levels differed markedly (Table 1). In summer 2015, 137 of 151 samples (91%) from singular beehives were contaminated with PA/PANO in the range of 0.2 to 111 µg/kg (Table 1; for single results see Appendix A). In spring 2016, 124 of the samples (82%) were contaminated with PA/PANO sum levels in the range of 0.1 to 59.4 µg/kg. In summer 2016, 134 samples were collected and 124 samples (93%) were contaminated with PA/PANO from 0.1 to 3313 µg/kg.

With respect to the distribution of the levels of contamination, the majority of the results of honey of summer 2015 (70%) were between >LOD and <10 µg/kg. Nineteen percent of the PA/PANO contents ranged between 10 and 50 µg/kg, while 2.0% (3 samples) showed results of >50 µg/kg (Figure 2). In spring 2016, a high percentage of the samples (82%) showed PA/PANO sum contents between LOD and <10 µg/kg. Only three samples were contaminated in the range of 10 to 50 µg/kg, one sample exceeded 50 µg/kg. Fifty-nine percent of the samples of summer 2016 showed levels between the LOD and 10 µg/kg. Twenty-four samples (18%) were contaminated in a range from 10 to 50 µg/kg and 19 samples (14%) contained levels of >50 µg/kg (Figure 2).

### 2.3. PA/PANO-Pattern and Botanical Origin

Regarding the identity of the analytes, typical *Senecio*-PA/PANO were detected in samples from every season (jacobine, senecionine, seneciphylline, retrorsine, erucifoline and their *N*-oxides as well as senkirkine). Furthermore, PA/PANO of other genera natively occurring in the area, such as *Echium*, *Eupatorium*, or *Borago,* were present (echimidine, intermedine, lycopsamine and their *N*-oxides) in a high percentage of the honey samples (Table 2). PA and their *N*-oxides of other Boraginaceae plants (europine, heliotrine, lasiocarpine) or of *Crotalaria* spp. (monocrotaline) were not detected in any sample (Appendix A).

Depending on the season, *Senecio*-type- or non-*Senecio*-type PA/PANO were detected with varying frequency. Within the samples of summer 2015, 12% exclusively contained lycopsamine-type PA/PANO, while 32% only contained *Senecio*-PA/PANO. In contrast, 98% of the samples of summer 2016 contained *Senecio*-PA/PANO. More than half of the positive samples of summer honeys, however, contained PA/PANO of various botanical origins. Unlike the honeys of summer harvest, PA/PANO detected in 45% of the spring samples exclusively originated from non-*Senecio* species, notwithstanding the fact that more than half of the samples were still contaminated with *Senecio*-PA (Table 2). 

When evaluating the contribution of *Senecio*- or non-*Senecio*-type PA/PANO to the total PA/PANO content, it turned out that the latter actually played a minor role. The maximum levels in all samples ranged between 2.8 and 6.4 µg/kg for lycopsamine and echimidine, respectively. In spring honey, echimidine and lycopsamine-type PA/PANO contributed to a higher extent, although their levels remained even lower than in summer honeys (Appendix A). 

With respect to the *Senecio* PA/PANO in honey of spring 2016, seneciphylline (31.0 µg/kg / 0.5 µg/kg), senecionine (16.5 µg/kg / 0.4 µg/kg) and jacobine (7.2 µg/kg / 0.1 µg/kg) were the main contributors to the PA/PANO sum (maximum level/mean of all samples). Slightly higher levels but with a similar pattern were found in the honey of summer 2015 with maximum and mean levels of 31.0 and 0.5 µg/kg for seneciphylline, 16.5 and 0.4 µg/kg for senecionine and 7.2 and 0.1 µg/kg for jacobine. The samples of both seasons contained comparably low levels of PANO (Table 3). On the contrary, in summer 2016, the levels of PANO were in the range of their corresponding PA or even higher. The highest mean or maximum levels ranged from 10.8 µg/kg for senecionine to 14.7 µg/kg for seneciphylline and from 482 µg/kg for senecionine to 711 µg/kg for senecionine *N*-oxide. In addition, jacobine and jacobine *N*-oxide showed mean and maximum levels of 3.6 µg/kg/504 µg/kg and 1.4 µg/kg/325 µg/kg, respectively. Moreover, erucifoline and retrosine and their *N*-oxides contributed markedly to the PA/PANO sum levels of summer honey 2016 (Table 3).

### 2.4. Inter- and Intra Site Variations

In summer 2015, spring 2016 and summer 2016 samples were, altogether, taken from 153 sites; some sites were visited in all three seasons, some sites were accessible only for one or two samplings. Regularly, the sampling of one site consisted of a set of three sampled beehives per site (subsamples A, B, C). This procedure allowed for comparing the PA/PANO contents of honey collected by individual bee colonies at the same site and for comparing seasonal influences.

The contents of PA/PANO varied markedly between the beehives of one site. In Table 4, the single results of mostly three beehives for all sampled sites with PA/PANO mean contents (mean of the samples A, B, C) of >10 µg/kg (*n* = 34) are displayed. It can be seen that singular bee colonies may be responsible for the contamination of the honey harvest of one site after mixing with the honey of other colonies, while other honeycombs of the same site were hardly contaminated. Between the lowest and the highest contaminated subsample, differences (expressed as Δ min-max in Table 4) of up to 800 µg/kg were observed. High relative standard deviations of the mean value of the three subsamples (up to 150%) underline the variance of PA/PANO contents between the different beehives. 

As a consequence, high PA/PANO contents are not necessarily dependent on the proximity to the PA/PANO-containing plants. As listed in the last column of Table 4, 19 of the 34 sites exhibiting mean levels of >10 µg/kg were in a distance of < 3 km to *J. vulgaris* (highly infested areas marked in Figure 3) which regularly is within the flight range of the honeybees. However, also honey collected from sites with a >3-km distance to areas with an enhanced growth of *J. vulgaris* (site 1a, 10a, 10b, 16a, 18a, 19c, 22c, 22d, 32a, 34a, 34b, 34c, 42a; Table 4, Figure 3) revealed PA/PANO contents of up to 150 µg/kg (site 1a, summer 2016, subsample C). 

The PA/PANO of these honeys were mostly all of *Senecio* origin (see percentages given in Table 4), including jacobine and erucifoline, giving a clear hint on *J. vulgaris* as their source. In honeys with a low PA/PANO contamination, a lower contribution of *Senecio*-type PA/PANO to the total content was also noticeable (sites 1f, 2b, 9b, each subsample C; sites 10 a, 15a, 16a, 22c, 34c, each subsample A; site 34 c, subsample B). However, some samples of summer honey 2016 from sites with proximity to *J. vulgaris* (sites 11a and 29a) also revealed a PA/PANO entry of non-*Senecio* origin of up to 16% of the total PA/PANO sum level (site 29a, subsample A).

### 2.5. Analysis of Pollen of Selected Samples

The results of the pollen analysis of each five samples per season are shown in Table 5. Pollen of relevance were detected from *Senecio*, *Eupatorium* and *Borago* spp. When evaluating the botanical origin of the PA/PANO, the majority of the selected samples contained metabolites of *Senecio* spp. (retrorsine, seneciphylline, senecionine, jacobine, erucifoline, and their *N*-oxides, and senkirkin). To a small extent, PA/PANO typical for *Eupatorium*, *Echium* or *Borago* species (intermedine, lycopsamine, echimidine, and their *N*-oxides) also contributed to the total PA/PANO sum content (results on single compounds see Appendix A). Regarding the distance to *J. vulgaris*, seven sites were within a 3-km range, eight sites were more distant than 3 km. As can be seen in Table 5, there was no clear connection with either the occurrence or absence of respective pollen nor the identity of measured PA/PANO. At most, the pollen of *Eupatorium* and *Borago* spp. may have been detected with a higher frequency from sites >3 km distance (Table 5).

The percentage of pollen of PA/PANO plants in all evaluated pollen (*n* = 500) was not higher than maximum 3.0% *Senecio*- (summer 2016, site 29b) or *Borago*-pollen (spring 2016, site 32a). Altogether, 108 different types of pollen were identified in all samples (results not shown). The presence of pollen did not necessarily indicate a presence of the respective analytes (e.g., summer 2015, site 16a) and high levels of PA/PANO did not imply a higher occurrence of pollen (e.g., summer 2016, site 1f). Regarding other samples of summer 2016, a certain correlation with the percentage of *Senecio*-pollen maybe seen in samples 34b, 29a, and 29b (Table 5).

### 2.6. Risk Assessment

Due to their acute hepatotoxic and carcinogenic properties exposure to PA/PANO from all food sources should generally be kept as low as possible. In contrast to retail honey, the consumption of regionally produced honey may result in a higher PA/PANO exposure, because that honey is not blended with honey from other sites or origins [39]. Therefore, the risks for consumers from the intake of PA/PANO via the honey of this study were assessed with respect to a chronic health risk due to longtime exposure (MOE evaluation) and to an acute health risk (HBGV evaluation). The calculations were conducted respecting all three seasons in a mean case scenario (applying the mean levels of contamination) and a worst-case scenario (applying the 95th percentile levels), based on the PA/PANO sum levels (medium-bound calculation) listed in Table 1. A worst-case scenario in this context actually reflects the situation in consumers of regionally produced and marketed honey as they usually have a high brand loyalty, like the beekeepers and their families themselves. 

Honey intake data for adults (60 kg b.w.) originated from the German National Nutrition Survey II. A mean and high consumption is quoted with 0.05 g/kg b.w./day and 0.28 g/kg b.w/day, respectively [49]. For 2- to 5-year-old children (16 kg b.w.), intake data from the German VELS-study were applied [50]. Average honey consumption by this sensitive subpopulation is quoted with 0.1 and 0.4 µg/kg b.w./day for mean and high consumers, respectively. For both consumer groups, values for an acute short-time honey intake were calculated with 0.88 and 1.36 g/kg b.w./day, which corresponds to approximately 52 g absolute daily honey intake for an adult and 22 g for children (95th percentile of intake of honey consumers). Precisely because the consumption habits of beekeepers and their families and customers can lead to a high daily intake over a long period of time, MOE values were also calculated for the acute consumption scenario. In addition, percentages of the HBGV were calculated in chronic exposure scenarios as well, as acute toxicity should also be assessed in view of cumulating PA/PANO intake from other sources.

For adults, neither mean nor high consumers of the honey of summer 2015 or spring 2016 harvest had worrying exposure levels. However, MOE values of <10,000 were calculated for consumers with a high and acute short-term intake of honey harvested in summer. Strikingly, consumers reflected by a worst-case scenario (consumers with a high brand loyalty) also faced a not negligible risk of exceeding the HBGV (Table 6).

For children, neither mean nor high consumers of the honey of summer 2015 and spring 2016 showed an elevated risk of PA/PANO exposure (Table 7). The calculations for the honey of summer 2016, however, reveal MOE levels below 10,000 already in the group of high consumers (laying down the mean level of contamination). High and acute intakes of summer honey 2016 in the mean and worst-case scenario of exposure resulted in percentages of the HBGV of 29% to 247%, which is clearly remarkable with regard to the fact that honey is only one source of daily PA/PANO exposure.

## 3. Discussion

All 437 honeys examined in the present study originated from a sampling area limited to a radius of about 20 km around Eutin, a district of Ostholstein (northern Germany). This area is characterized on the one hand by a high number of beekeepers and on the other hand by nature reserves with an immense distribution of *J. vulgaris* (Figure 1). Samples of raw, unblended honey were directly taken from individual beehives and examined for their PA/PANO content. The aim of this study was to obtain a detailed picture of the origin and extent of the PA/PANO contamination in honeys from a regionally limited area and then to conduct a risk assessment for consumers of regionally produced honey. As far as we know, an investigation of a comparably large number of samples from individual bee colonies at a regionally restricted area has not yet been carried out. Similar studies with raw or regionally produced honeys (see Appendix A) reported on the contents of honeys of individual locations or floral varieties, but these data did not allow for tracing back the results to individual bee colonies at the individual locations. In three studies, sampling directly from the honeycombs was performed as well, but with a smaller sample size [3,21,25].

In summer honey 2016, this study revealed a maximum contamination of 3313 µg/kg of a site, which was within a distance of 3 km to *J. vulgaris*. This radius can be regarded as an average foraging area of honeybees, which can be lower or even higher depending on the landscape, foraging month, nectar supply, and bee species [51,52,53]. The PA/PANO sum level was comparable to early studies of Deinzer et al. (1977) and Crews et al. (1997), where honey was experimentally sourced in proximity to *J. vulgaris* [16,18]. Moreover, other studies summarized in Appendix A reported a comparable entry of PA/PANO into honey from known botanical sources: Experimental raw honeys from Boraginaceae plants sources (*Echium* and *Heliotropium* spp.) showed maximum values of up to 2634 µg/kg in Australian honey [17,19]. In raw honey originating from Central or South America PA/PANO sum levels of up to 1087 µg/kg were reported [20]. Neumann et al. (2016) conducted a study with 86 summer honeys from the same region. The honeys harvested in 2014 revealed a mean level of contamination of 34 µg/kg and a maximum level of 604 µg/kg, which was below the levels reported in our study [26]. Both studies included a similar range of recorded PA/PANO compounds (LC–MS/MS target analysis of 28 or 25 individual compounds). The mean contents of the summer honey 2015 and spring honey 2016 of the current study (8.4 and 1.5 µg/kg, respectively) were in a similar range as those of regionally produced honeys from Poland (4.5 µg/kg) [28], Switzerland (3.6 µg/kg) [23] or Germany and Austria (6.1 µg/kg) [24]. The mean of 26 µg/kg determined for raw honeys from the EU [20] and the mean value of the study by Neumann et al. 2016 [26] was about one third or half of the mean level of the summer honeys 2016 from this study, respectively.

Another study that is quite comparable to ours in terms of geography and botanical origin was conducted in the Netherlands by Kempf et al. [21]. However, raw honeys (*n* = 31) sourced in proximity to *J. vulgaris* revealed a mean and maximum level (quantified as retronecine equivalents) of 1261 and 13,019 µg/kg. These results suggested that the analytical method (recording all PA/PANO of a common retronecine body after chemical derivatization) included a larger number of compounds not being part of target analyses [21]. Due to the back then limited availability of reference standards, the measurements of the samples included in this study were restricted to a spectrum of 25 PA/PANO. Another recent comparison of results from both approaches also revealed differences in the levels of the determined PA/PANO sum- or retronecine equivalent levels for certain samples [21,54]. In the meantime, our method has been amended by 31 PA/PANO compounds, which comprise further metabolites of *Senecio* spp., such as integerrimine, jaconine, and riddelliine as well as isomers of lycopsamine. Recent analyses of *J. vulgaris* plants originating from the same district as that of our study showed those compounds to contribute markedly to the total PA/PANO sum content of this plant (unpublished results). Other recent studies described 98 PA/PANO compounds occurring in *J. vulgaris* or even 176 PA/PANO occurring in Boraginaceae plants which have been harvested in the same region in northern Germany [55,56]. Therefore, it is highly likely that the measurement results of this study and previous target analyses—also if the number of included compounds constantly increased (Appendix A)—are subject to underestimation. With respect to the uncertainties of the analytical scope on exposure calculations of PA/PANO [4,39,56,57], an expansion of the future legislation covering 35 PA/PANO is currently under discussion [56]. 

Regarding further uncertainties and influences, plant-specific and seasonal fluctuations in the PA/PANO levels must also be taken into account. A recent study of Flade et al. showed clear differences in the PA and PANO contents depending on the season and stage of the development of the plant (*S. vulgaris*) [58]. *S. aquaticus* as well as *S. inaequidens* also showed high seasonal fluctuations in PA/PANO occurrences [59,60]. Among other reasons, this could explain differences in the levels of the summer seasons (Table 1) and between different studies [26]. Furthermore, the stability of PANO in honey was recently examined in natural and artificially contaminated honeys. It appeared that, during storage, PANO decreased rapidly within the period of evaluation (8 weeks) and, therefore, the comparability of results is strongly influenced by the time of analysis after honey harvest [54,61]. Similar observations on PANO decrease were recently reported for bee bread [62]. It is still unknown whether this decrease actually corresponds to a detoxification or whether products still incorporating a toxic potential were formed. At present, it remains to be discussed whether the PA/PANO exposure and thus the risk of honey consumption may depend on the age or time of storage of the respective product. As far as the results from summer 2015 and spring 2016 of our study are concerned, the PANO contents of these samples reflected the findings on the decrease of PANO as described before. The individual results (Table 3) showed that PANO were hardly detectable in these seasons. In contrast to the samples from summer of 2016, these samples were not directly examined after harvest, but stored up to two months before analysis. The samples of summer 2016 were analyzed shortly after harvest and PANO mostly contributed equally to the PA/PANO sum content like the corresponding tertiary amines.

The detection of a high percentage of PA/PANO not originating from *Senecio* (Table 2) highlights the role of other plant species, which contribute to the allover PA/PANO content in honey of this region. There are not only *J. vulgaris* but also *Eupatorium cannabinum* or Boraginaceae plants which could lead to a PA/PANO contamination due to a common distribution of these plants [63]. However, *Senecio* PA/PANO mostly contributed to the allover PA/PANO sum content, which was also observed earlier [26]. A high frequency of non-*Senecio* PA/PANO was observed especially in the spring honeys, where nearly half of the contaminated samples only contained non-*Senecio* PA/PANO. At this time, the bees still have a wide range of alternative plants, which explains why the proportion of PA/PANO from the *Senecio* spectrum in spring is lower than in summer harvests. *J. vulgaris* gains in importance at the end of July at the peak of their flowering period and offer bees an alternative opportunity when the nectar supply of other plants becomes scarce [64]. The earlier flowering time of some plants of the Boraginaceae family may be responsible for the more frequent occurrence of PA/PANO of the lycopsamine type. A correlation with the detected pollen may show a tendency towards this (Table 5), since *Borago* pollen were detected only in samples from spring 2016. At the same time, the *Senecio* pollen content was consistently lower than in the honeys of the two summer seasons. However, Kast et al. have shown for *Echium* spp. in Switzerland that bees collected only a few pollen of this plant and concluded that a monitoring of pollen loads was not suited to estimate a risk of PA/PANO contamination [62]. A Spanish study performed by Orantes-Bermejo et al. also reported a non-correlation between presence of *Echium* pollen and PA/PANO levels in bulk honey samples [22]. This could be a reason why we had echimidine-positive samples, but no *Echium* pollen were detected in the selected samples of our study (Table 5, Appendix A). 

The results of the samples originating from various sites being near (<3 km) or far from *J. vulgaris* (Figure 3, Table 5) proved that levels in the mg/kg range could result from beehives in proximity to these toxic plants. However, it was clearly shown as well that beehives in proximity to *J. vulgaris* had also been sampled, which revealed very low PA/PANO levels (e.g., site 1f, 10b, 11a, 19a, 22c, 32a), while others at the same site were highly contaminated. Differences in the range of 50 to 300 µg/kg were not uncommon between the three subsamples A, B and C (of the three sampled beehives per site), up to a difference of over 800 µg/kg at the site with the highest contamination in summer 2016 (site 35a, Figure 3). In view of our results of a PA/PANO contamination of up to 150 µg/kg at sites with a distance of >3 km to known occurrences of *J. vulgaris* (e.g., site 1a, 1c, 34b) and the fact that for bees even a flight radius of up to 5 km is reported [53], it seems practically impossible for a beekeeper to guarantee the absence of undesirable plants. A contamination of the entire harvest could be avoided by testing honey from single beehives prior to mixing with others. Instead of complex and expensive LC–MS/MS procedures, cost-effective immunoassays that can be used on site would be suitable for this purpose [65]. However, these are currently not yet commercially available.

The health risk assessment calculations revealed that, depending on the season and the applied scenario of intake, consumers—and particularly the subpopulation of children – face a low or a high risk of chronic as well as of acute adverse health effects (Table 6 and Table 7). The hazard characterization corresponded to the currently applied values of the responsible authorities [4,41]. However, the risk assessment resulted in partly worrying results due to the levels of occurrence of PA/PANO (Table 1) and the intake scenarios of honey. Regarding the measured PA/PANO levels that were used to calculate the exposure, resilient data about the affected region can be assumed due to the dispersion on various sites and the high number of individual samples (*n* = 151, 152, and 134 for summer 2015, spring 2016, and summer 2016, respectively). 

As can be seen in Table 6 and Table 7, the average consumer (subpopulation of adults and children) did not face a risk of PA/PANO exposure via the consumption of contaminated honey neither in a mean-case (mean level of PA/PANO contamination) nor a worst-case (95th percentile level of contamination) scenario of both spring and summer honey. However, in our opinion, consumers of regionally produced honey specifically select honey from a singular beekeeper (i.e., a high brand loyalty)—differing from the average population. In particular, beekeepers and their families as well as their customers are regularly subject to a worst-case scenario of exposure because it is unlikely that they consume retail honey or products from other origins. Additionally, it is more than likely that their honey consumption is higher than the average amounts of only 3 g/day (adults) or 1.6 g (children) as quoted in German official nutrition surveys [49,50]. The acute short-time intakes (Table 6 and Table 7) correspond to 52 g of absolute daily honey intake for an adult and 22 g for children. Moreover, a recent Australian study performed by Carpinelli de Jesus et al. referred to children of 2–4 years having a honey intake of up to 28.6 g/day [29]. In consideration of the size of a regular honey serving (25-g-portion pack) and of an amount of 40 g of honey, which are regularly spread on three average slices of bread, the used high and acute daily consumption scenarios are quite realistic for the above-mentioned consumer groups. Nevertheless, this is a long-term consumption habit rather than a short-time scenario so that we have conducted MOE assessments for this scenario as well. Consequently, summer honey consumption in high and acutely consuming adults may result in an exposure below the MOE of 10,000 and even slightly exceed the HBGV of 0.1 µg/kg of body weight. However, this was much more pronounced in children (2 to 5 years of age) who showed an MOE of less than 10,000 at high consumption already at the assumption of the mean PA/PANO contamination levels (Table 7). Hereby, the applied exposure scenarios are quite similar to the risk assessment of honey of the BfR of 2011 based on a mean and 95th percentile level of 31.5 and 74.7 µg/kg, respectively [39]. An acute consumption, which appeared—as mentioned before—quite probable in this special group of consumers, resulted in an exceedance of the HBGV by more than 2-fold.

Regarding current German risk management activities (based on an MOE 10,000 of 0.0237 µg/kg b.w./day [4] and the average consumption of 3 g honey per day (adult, 60 kg b.w.) [49]), a guidance value of 474 µg/kg honey was suggested for application within the official food surveillance [66]. However, this value neither respects an intake from further sources (tea, spices, food supplements) nor the sensitive subpopulation of children (16 kg b.w.). In fact, the results of the risk assessment showed that high-consuming children (6.5 g honey per day) already had MOE values of <10,000 when consuming the honey of summer 2016 with an average contamination (Table 7). As a result, a guidance value should be significantly lower in order to protect children adequately from these carcinogenic contaminants. In general, the implementation of maximum levels for PA/PANO in certain foodstuffs to Commission Regulation No 1881/2006 and the scope of PA/PANO included into the respective sum level is still under discussion. This does not, however, apply to honey [56,67].

As shown by the results of this study, honeys can endanger the consumer if PA/PANO-forming plants are predominant in the region and a virtually safe distance (>3 km) to toxic plants did not guarantee safe honey. It has been shown that the PA levels of beehives at one site can vary greatly. Therefore, highly contaminated honey from a single bee colony could contaminate an entire batch. Considering the number of approximately 150,000 beekeepers in Germany [68], their families, and consumers of regionally produced honeys, it can be expected that more than 1 million consumers could face an elevated risk of PA/PANO exposure, because they face a worst-case scenario due to high brand loyalty and higher intake. Additionally, it must be considered that the consumers’ interest in regionally produced and marketed products is increasing. This also applies to so-called ‘city honeys’, for which the botanical source of the honey can hardly be controlled at all [29]. Consequently, it is highly recommendable to adequately respect regionally produced honey as a special source of PA/PANO and to identify beekeepers and consumers of regionally produced honey as a subpopulation with a higher exposure. Therefore, exposure and risk assessments should not only cover the average consumer of mixed retail honeys.

## 4. Materials and Methods 

### 4.1. Samples

Sampling was performed in cooperation with 42 beekeepers from the district of Ostholstein, Germany. The samples were directly taken from singular beehives at 73 different locations shortly before the regular honey harvest, i.e., the end of July 2015, the end of May 2016, and the end of July 2016. During these three consecutive seasons of sampling (here referred to as “summer 2015”, “spring 2016”, “summer 2016”; Table 1), some sites were sampled in all seasons, others only in one or two (Appendix A). Exact geographical data were monitored for every sample and used for the evaluations in this study; however, they are not provided due to data protection requirements. The honeycombs were taken out and raw honey (approximately 200 g) was taken off by using a wooden spatula. In order to evaluate variations in PA/PANO levels, usually three beehives were sampled at each site (subsamples A/B/C). Altogether, 437 samples were collected (Table 1). Until analysis, the samples were stored at 4 °C.

### 4.2. Chemicals

Reference substances for 25 PA and PANO were purchased from Phytolab, Vestenbergsgreuth, Germany (echimidine, echimidine *N*-oxide, europine, europine *N*-oxide, erucifoline, erucifoline *N*-oxide, heliotrine, heliotrine *N*-oxide, intermedine, intermedine *N*-oxide, jacobine, jacobine *N*-oxide, lasiocarpine, lasiocarpine *N*-oxide, lycopsamine, lycopsamine *N*-oxide, monocrotaline, monocrotaline *N*-oxide, retrorsine, retrorsine *N*-oxide, senecionine, senecionine *N*-oxide, seneciphylline, seneciphylline *N*-oxide, and senkirkine). Stock solutions (c = 1.0 mg/mL) of each PA/PANO were prepared either with acetonitrile or acetonitrile/water (50/50, v/v), depending on the solubility, and stored at 4 °C in the dark. A PA/PANO mix solution (c = 1.0 µg/mL of each analyte) was used to prepare calibration standards and for the validation experiments. Water, acetonitrile and methanol were used in LC–MS grade (Th. Geyer, Renningen, Germany). Ammonia solution (32%) was purchased from Th. Geyer as well. Formic acid and ammonium formate used as eluent additive (LC–MS/MS grade) were from Sigma Aldrich (Deisenhofen, Germany). 

### 4.3. Sample Pretreatment

Honey samples were placed in a water bath at 38 °C and were homogenized by using a spatula. If necessary, wax particles or impurities were removed by filtering through a sieve (pore size 1.5 mm). The method for extraction and sample clean-up was based on a previously published procedure [24]. An aliquot of each honey sample (10 g) was weighed into a 50-mL tube, diluted in 30 mL sulfuric acid (0.05 mol/L) and placed onto a horizontal shaker (30 min, 300 rpm). Afterwards, the samples were centrifuged (20 min, 20 °C, 2600× *g*). The supernatant was completely passed through a HF BondElut LRC SCX column (500 mg/6 mL) from Agilent (Waldbronn, Germany), preconditioned and equilibrated with each 5 mL of methanol and sulfuric acid (0.05 mol/L). The columns were washed with 6 mL each of water and methanol (LC–MS grade) and the analytes were eluted with 10 mL of ammonia (2.5%) in methanol. The extracts were evaporated to dryness under a gentle stream of nitrogen at 50 °C and reconstituted in 1.0 mL methanol (10%) by means of ultrasonication (1 min) and a laboratory mixer. Prior to analysis, the extracts were filtered through a syringe filter (PVDF membrane, 0.45 µm) into a HPLC vial.

### 4.4. LC–MS/MS Measurement and Quantification

Measurements were carried out using a Shimadzu Prominence HPLC-system consisting of a binary pump LC-20AB, an autosampler SIL-20AC HT, a column oven CTO-20AC and a controller unit CBM-20A (Duisburg, Germany) and an API4000 triple quadrupole mass spectrometer (Sciex, Framingham, MA, U.S.A). For chromatographic separation, a Synergi™ RP-polar analytical column (150 × 2.0 mm, 4 µm) from Phenomenex (Darmstadt, Germany) was used. Solvents A and B consisted of water or methanol, respectively, both containing ammonium formate (5 mmol/L) and formic acid (0.1%). A binary linear gradient with the following parameters was used: 0 min, 5% B; 5 min, 5% B; 10 min, 25% B; 14 min, 25% B; 22 min, 95% B; 22.1 min, 5% B; 28 min, 5% B. The injection volume was 20 µL and the oven temperature was set at 40 °C.

The multiple reaction monitoring (MRM) transitions and substance-specific parameters for 25 detected PA/PANO are provided as Appendix A. The ion source parameters were set as follows: temperature, 650 °C; ionization voltage, 2500 V (ESI +); nebulizer gas 50 psi; heating gas, 50 psi; curtain gas, 30 psi; collision gas level, 7. Sciex Analyst (Version 1.6.2, Framingham, MA, U.S.A) and MultiQuant software (Version 3.0.1, Framingham, MA, U.S.A) were used for data acquisition and evaluation.

The analytes were quantified by matrix-assisted calibration. Therefore, aliquots of the PA/PANO mix solution were pipetted into HPLC vials and evaporated under a gentle stream of nitrogen at 50 °C. The standards were reconstituted in 1 mL of an extract of a blank honey sample, which was prepared as described above. The final concentrations were 1.0, 5.0, 10.0, 25.0, and 50.0 ng/mL. For calculations, the results <LOD were set to “0 µg/kg” and results <LOQ were regarded as 0.5 LOQ (medium bound calculation, [39]). The results were not corrected for the analytical recovery rate.

### 4.5. Method Validation

The established method was inhouse validated by assessing the analytical recovery, precision and linearity. Samples of a blank honey were fortified with the PA/PANO mixture at three levels (5.0, 25 and 50 µg/kg) with three repetitions each and were prepared as described above. The limit of detection and quantification (LOD, LOQ) and the linearity was assessed by the linear regression/calibration curve method according to German DIN 32645 [69] using matrix-matched calibration standards in the range of 0.01–100 ng/mL.

### 4.6. Pollen Analysis

Pollen analysis of exemplarily selected samples (each = 5 of every season) was performed by the Lower Saxony State Office for Consumer Protection and Food Safety (LAVES Niedersachsen, Germany). Prior to taking out an aliquot for shipment, the samples were thoroughly homogenized as described above. Each *n* = 500 pollen was differentiated per sample, and the percentage of pollen originating from relevant species (*Senecio*, *Eupatorium*, *Echium*, *Borago*) was calculated. The analyses aimed at examining the plant sources of PA/PANO contamination and in comparing with measured PA/PANO levels. 

## Figures and Tables

**Figure 1 toxins-12-00441-f001:**
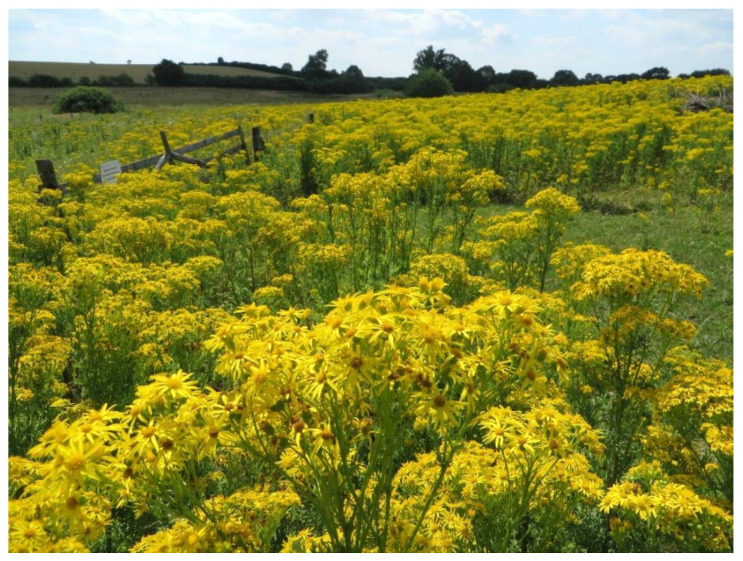
Spread of *Jacobaea vulgaris* (tansy ragwort) on nature conservation areas near the city of Eutin, district of Ostholstein, Germany.

**Figure 2 toxins-12-00441-f002:**
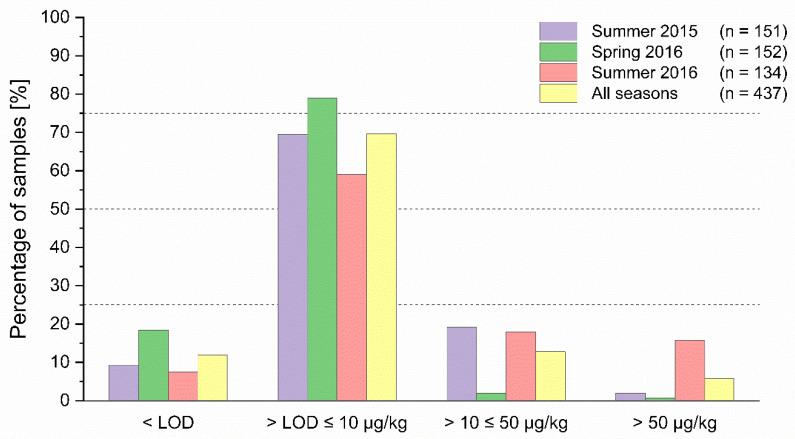
Comparison of the frequency of the detection and distribution of pyrrolizidine alkaloids (PA) and PA *N*-oxides (PANO) sum levels in honey depending on the season.

**Figure 3 toxins-12-00441-f003:**
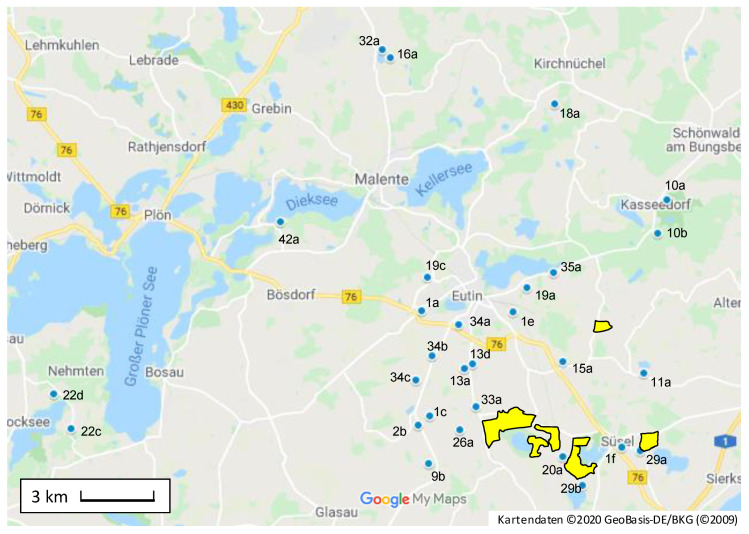
Blue spots mark selected sites of sampling in the area of Eutin, Ostholstein, Germany, with pyrrolizidine alkaloid (PA) and PA *N*-oxide (PANO) sum levels of >10 µg/kg (mean level of the beehive subsamples A, B, and C, see also Table 5). Yellow marked areas show areas with heavy infestation with *Jacobaea vulgaris* in the south of Eutin. Designations indicate beekeeper and site.

**Table 1 toxins-12-00441-t001:** Specifications and results (sum of pyrrolizidine alkaloids (PA) and PA *N*-oxides) on honey samples (*n* = 437) collected from single beehives at various sites and from three consecutive seasons.

Season		N		Incidence (%)	PA/PANO Sum Content (µg/kg)			
	Bee- Keepers	Sites	Samples		Median	Mean	95th Percentile	Maximum
Summer 2015	30	53	151	91	2.3	8.4	35.4	111
Spring 2016	35	53	152	82	0.2	1.5	3.6	59.4
Summer 2016	29	47	134	93	3.9	72.6	181	3313
All	42	73	437	88	1.0	25.7	59.5	3313

**Table 2 toxins-12-00441-t002:** Percentage of samples exclusively contaminated with pyrrolizidine alkaloids (PA) and PA *N*-oxides (PANO) originating from *Senecio* or from *Borago/Echium/Eupatorium* species and samples containing mixed botanical origins in three different seasons.

Season	*Senecio*-Type	*Borago*/*Echium*/*Eupatorium*-Type	Mixed Origin
Summer 2015	32%	12%	56%
Spring 2016	26%	45%	29%
Summer 2016	44%	2.0%	54%

**Table 3 toxins-12-00441-t003:** Single pyrrolizidine alkaloid (PA)/PA *N*-oxide (PANO) compounds of the *Senecio*-type detected in the honey samples of the seasons of summer 2015, spring 2016, and summer 2016.

Season (*n* Samples)		PA/PANO (*Senecio*-Type) (µg/kg)										
	Incidence (%)/Levels	Ec	EcN	Jb	JbN	Rs	RsN	Sc	ScN	Sp	SpN	Sk
Summer 2015 (*n* = 151)	incidence	40	7.9	60	15	56	5.3	70	21	66	25	7.9
	mean	0.1	0.0	1.2	0.0	0.4	0.0	2.8	0.1	3.3	0.1	0.0
	95th perc.	0.6	0.3	5.5	0.2	1.8	0.1	15.0	0.5	15.3	0.5	0.1
	max.	1.9	0.3	16.2	0.3	4.6	0.4	40.4	2.0	40.7	2.6	0.7
Spring 2016 (*n* = 152)	incidence	3.3	0.7	13	11	18	3.3	23	32	22	29	7.9
	mean	0.0	0.0	0.1	0.0	0.1	0.0	0.4	0.1	0.5	0.1	0.0
	95th perc.	0.0	0.0	0.3	0.1	0.2	0.0	1.1	0.3	1.1	0.4	0.0
	max.	0.7	0.3	7.2	1.4	2.7	0.7	16.5	3.8	31.0	4.9	0.2
Summer 2016 (*n* = 134)	incidence	29	21	64	54	49	40	89	73	84	77	12
	mean	0.7	2.1	10.3	4.4	2.4	1.7	10.8	12.7	14.7	12.1	0.0
	95th perc.	5.0	7.4	22.1	10.6	4.0	4.9	33.1	36.3	38.3	40.5	0.2
	max.	16.4	111	504	325	136	120	482	711	609	630	0.3

95th perc., 95th percentile; Ec, erucifoline; EcN, erucifoline *N*-oxide; Jb, jacobine; JbN, jacobine *N*-oxide; Rs, retrorsine; RsN, retrorsine *N*-oxide; Sc, senecionine; ScN, senecionine *N*-oxide; Sp, seneciphylline; SpN, seneciphylline *N*-oxide; Sk, senkirkine.

**Table 4 toxins-12-00441-t004:** Results (sum of pyrrolizidine alkaloids (PA) and PA *N*-oxides (PANO)) for different beehives at one sampled site (subsamples named A, B, or C) from different seasons, including the identity of measured PA/PANO (percentage *Senecio*-type). Only samples of PA/PANO mean contents of >10 µg/kg are shown (the complete results are available as Appendix A). Furthermore, the distance of each site to areas with an enhanced spread of *Jacobaea vulgaris* is specified (<3 km = usual harvesting range of honeybees).

Site- ID	Season	PA/PANO Sum Content (µg/kg) (Percentage *Senecio*-Type)			Mean (µg/kg)	RSD (%)	Δ min-max (µg/kg)	Distance *J. vulgaris*
		Sample A	Sample B	Sample C				
1a	spring 2016	22.4 (100%)	0.9 (100%)	59.4 (100%)	27.6	107	58.5	>3 km
1a	summer 2016	18.8 (100%)	130 (100%)	151 (99.9%)	99.8	71	132	>3 km
1c	summer 2016	9.5 (100%)	45.9 (100%)	135 (100%)	63.4	102	125	<3 km
1e	summer 2016	38.8 (100%)	60.4 (100%)	20.9 (100%)	40.0	49	39.5	<3 km
1f	summer 2016	346 (100%)	26.5 (100%)	1.3 (15.1%)	125	154	345	<3 km
2b	summer 2015	17.4 (96.1%)	41.9 (98.3%)	8.6 (93.0%)	22.7	76	33.3	<3 km
9b	summer 2015	30.9 (97.5%)	30.1 (99.3%)	4.7 (93.5%)	21.9	68	26.2	<3 km
10a	summer 2015	3.3 (93.1%)	37.0 (98.6%)	33.3 (99.6%)	24.6	75	30.0	>3 km
10b	summer 2015	33.8 (99.6%)	6.4 (100%)	0.5 (100%)	13.6	131	33.3	>3 km
11a	summer 2016	33.0 (90.2%)	22.1 (94.2%)	1.3 (100%)	18.8	86	31.7	<3 km
13a	spring 2016	7.2 (100%)	1.0 (97.4%)	24.8 (100%)	11.0	112	23.8	<3 km
13d	summer 2016	1.1 (100%)	28.4 (100%)	5.1 (100%)	11.5	128	23.3	<3 km
15a	summer 2016	7.7 (72.9%)	16.2 (96.5%)	35.2 (95.3%)	19.7	72	27.5	<3 km
16a	summer 2015	3.9 (91.8%)	79.5 (100%)	n.a.	41.7	128	75.6	>3 km
18a	summer 2015	7.1 (100%)	41.8 (97.6%)	5.3 (100%)	18.0	114	36.5	>3 km
19a	summer 2016	25.1 (100%)	19.9 (100%)	0.4 (100%)	15.2	86	24.7	<3 km
19c	summer 2016	2.5 (100%)	34.5 (100%)	6.1 (100%)	14.4	122	32.0	>3 km
20a	summer 2015	24.3 (100%)	27.8 (99.5%)	21.6 (100%)	24.6	13	6.2	<3 km
20a	summer 2016	12.7 (98.2%)	47.8 (100%)	331 (100%)	131	134	319	<3 km
22c	summer 2015	0.4 (46.1%)	47.6 (99.7%)	3.9 (96.4%)	17.3	152	47.2	>3 km
22d	summer 2015	111 (98.7%)	21.2 (95.3%)	5.5 (97.3%)	45.7	124	105	>3 km
26a	summer 2015	22.5 (99.0%)	4.4 (92.8%)	5.7 (94.9%)	10.9	92	18.1	<3 km
29a	summer 2015	27.2 (99.2%)	10.4 (100%)	n.a.	18.8	63	16.8	<3 km
29a	summer 2016	64.2 (84.0%)	101 (95.7%)	314 (95.4%)	160	84	249	<3 km
29b	summer 2015	29.5 (100%)	58.1 (99.0%)	32.2 (100%)	39.9	40	28.6	<3 km
29b	summer 2016	173 (98.9%)	446 (99.7%)	197 (99.3%)	272	56	273	<3 km
32a	spring 2016	9.0 (100%)	39.2 (99.2%)	0.6 (100%)	16.3	125	30.2	>3 km
33a	summer 2015	12.0 (94.0%)	44.9 (99.5%)	22.0 (100%)	26.3	64	32.9	<3 km
34a	summer 2016	48.3 (100%)	25.7 (100%)	20.2 (99.6%)	31.4	47	28.1	>3 km
34b	summer 2016	61.9 (100%)	133 (100%)	136 (99.8%)	110	38	74.2	>3 km
34c	summer 2015	6.5 (94.3%)	2.4 (73.1%)	29.3 (98.0%)	12.7	114	26.9	>3 km
34c	summer 2016	79.5 (100%)	81.8 (100%)	59.7 (100%)	73.7	17	22.1	>3 km
35a	summer 2016	3313 (99.9%)	2510 (99.9%)	n.a.	2912	20	803	<3 km
42a	summer 2016	53.0 (98.4%)	21.2 (95.0%)	31.5 (99.9%)	35.2	46	31.8	>3 km

n.a. = not analyzed; RSD = relative standard deviation; Δ min-max = difference between subsample with highest and lowest PA/PANO sum content.

**Table 5 toxins-12-00441-t005:** Results of pollen analysis in each *n* = 5 samples per season in comparison with pyrrolizidine alkaloid (PA)/PA N-oxide (PANO) levels and botanical origin of detected PA/PANO.

Season	Site- ID	Sample ^1^	PA/PANO		Percentage (%) of Pollen ^2^			Distance *J. vulgaris*
			Sum Level (µg/kg)	*Senecio*-Type (%)	*Senecio*	*Eupatorium*	*Borago*	
Summer 2015	16a	B	79.5	100	2.0	2.0	n.d.	>3 km
	20a	B	27.8	99.5	n.d.	n.d.	n.d.	<3 km
	22d	A	111	98.7	<0.1	n.d.	n.d.	>3 km
	29b	B	58.1	99.0	2.0	<0.1	n.d.	<3 km
	33a	B	44.9	99.5	n.d.	n.d.	n.d.	<3 km
Spring 2016	1a	C	59.4	100	<0.1	<0.1	n.d.	>3 km
	8a	B	2.4	0.0	<0.1	n.d.	n.d.	>3 km
	15a	B	7.5	100	<0.1	n.d.	n.d.	<3 km
	21a	A	5.2	93.2	<0.1	n.d.	n.d.	>3 km
	32a	B	39.2	99.2	n.d.	n.d.	3.0	>3 km
Summer 2016	1f	A	346	100	<0.1	n.d.	n.d.	<3 km
	22a	A	6.7	100	<0.1	n.d.	n.d.	>3 km
	34b	C	136	99.8	1.0	0.5	n.d.	>3 km
	29a	C	314	95.4	2.0	1.0	n.d.	<3 km
	29b	B	446	99.7	3.0	n.d.	n.d.	<3 km

^1^ Three beehives (subsamples A/B/C) were sampled per site; ^2^ The pollen of relevance to PA/PANO-producing plants found in the selected samples. *Echium* pollen were not identified. The complete results on the identity of each differentiated pollen (*n* = 500, altogether 108 different types) are available from the authors upon request; n.d. = not detected.

**Table 6 toxins-12-00441-t006:** Calculation of pyrrolizidine alkaloid (PA) and PA *N*-oxide (PANO) exposure and of the chronic and acute health risks for adults (60-kg body weight). Values of concern are marked in bold.

Season	Intake ^1^ (g honey/kg b.w./day)	Exposure Scenario ^2^	PA/PANO Intake (ng/kg b.w./day)	MOE ^3^	Percentage of HBGV ^4^
summer 2015	mean (0.05)	mean case	0.42	564,000	0.4%
		worst case	1.8	134,000	1.8%
	high (0.28)	mean case	2.4	101,000	2.4%
		worst case	9.9	23,900	9.9%
	acute (0.88)	mean case	7.4	32,100	7.4%
		worst case	31.2	**7610**	31.2%
spring 2016	mean (0.05)	mean case	0.08	3,160,000	0.1%
		worst case	0.18	1,317,000	0.2%
	high (0.28)	mean case	0.42	564,000	0.4%
		worst case	1.0	235,000	1.0%
	acute (0.88)	mean case	1.3	180,000	1.3%
		worst case	3.2	74,800	3.2%
summer 2016	mean (0.05)	mean case	3.6	65,300	3.6%
		worst case	9.1	26,100	9.1%
	high (0.28)	mean case	20.3	11,700	20.3%
		worst case	50.8	**4670**	50.8%
	acute (0.88)	mean case	63.9	**3710**	63.9%
		worst case	160	**1490**	**160%**

^1^ Intake data from German National Nutrition survey II [49], b.w. = body weight; ^2^ Levels of contamination as listed in Table 1. Mean PA/PANO sum level = mean case; 95th percentile level = worst case; levels < LOQ were set to 0.5 LOQ (medium bound calculation); ^3^ MOE = margin of exposure. Calculation based on a benchmark dose lower confidence limit 10% (BMDL_10_) of 237 µg/kg b.w./day (chronic health risk) [4,43]; ^4^ calculation based on a health-based guidance value (HBGV) of 0.1 µg/kg b.w./day (acute health risk) [41,45]. Values of concern are marked in bold.

**Table 7 toxins-12-00441-t007:** Calculation of pyrrolizidine alkaloid (PA) and PA *N*-oxide (PANO) exposure and of the chronic and acute health risks for 2 to 5-years aged children (16-kg body weight). Values of concern are marked in bold.

Season	Intake ^1^ (g honey/kg b.w./day)	Exposure Scenario ^2^	PA/PANO Intake (ng/kg b.w./day)	MOE ^3^	Percentage of HBGV ^4^
summer 2015	mean (0.10)	mean case	0.84	282,000	0.8%
		worst case	3.5	66,900	3.5%
	high (0.40)	mean case	3.4	70,500	3.4%
		worst case	14.2	16,700	14.2%
	acute (1.36)	mean case	11.4	20,700	11.4%
		worst case	48.1	**4920**	48.1%
spring 2016	mean (0.10)	mean case	0.15	1,580,000	0.2%
		worst case	0.36	658,000	0.4%
	high (0.40)	mean case	0.60	395,000	0.6%
		worst case	1.4	164,000	1.4%
	acute (1.36)	mean case	2.0	116,000	2.0%
		worst case	4.9	48,400	4.9%
summer 2016	mean (0.10)	mean case	7.3	32,600	7.3%
		worst case	18.1	13,100	18.1%
	high (0.40)	mean case	29.0	**8160**	29.0%
		worst case	72.5	**3270**	72.5%
	acute (1.36)	mean case	98.7	**2400**	98.7%
		worst case	247	**960**	**247%**

^1^ Intake data from VELS-study [50], b.w. = body weight; ^2^ Levels of contamination as listed in Table 1. Mean PA/PANO sum level = mean case; 95th percentile level = worst case; levels < LOQ were set to 0.5 LOQ (medium bound calculation); ^3^ MOE = margin of exposure. Calculation based on a benchmark dose lower confidence limit 10% (BMDL_10_) of 237 µg/kg b.w./day (chronic health risk) [4,43]; ^4^ Calculation based on a health-based guidance value (HBGV) of 0.1 µg/kg b.w./day (acute health risk) [41,45]. Values of concern are marked in bold.

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
