# Peer review of "Spread of *Jacobaea vulgaris* and Occurrence of Pyrrolizidine Alkaloids in Regionally Produced Honeys from Northern Germany: Inter- and Intra-Site Variations and Risk Assessment for Special Consumer Groups"

_toxins, 2020, doi:10.3390/toxins12070441_

Round 1

Reviewer 1 Report

The study is overall interesting. The findings can provide useful information about why it is important to identify regionally produced honey as a special source of PA/PANO and also to identify beekeepers and consumers of regionally produced honey as a subpopulation with higher exposure to PA/PANO contamination.

In my opinion, the introduction provides sufficient background, the research design is appropriate the methods are adequately described and the results are clearly presented.

Authors should also mention the name of the first author for some of the studies mentioned in the Discussion example: “Another study … conducted in the Netherlands by X et al.”, “A Spanish study performed by Y et al.”, etc.

Conclusions should be better highlighted; they seem to be lost in the Discussion.

Author Response

Thanks for reviewing this manuscript and giving important remarks. As requested we amended some sentences in the discussion by the first authors of the respective studies:

-Another study that is quite comparable to ours in terms of geography and botanical origin was conducted in the Netherlands by Kempf et al. [21]. 

-A recent study of Flade et al. showed clear differences in the PA and PANO contents depending on the season and stage of development of the plant (S. vulgaris) [58]. 

-A Spanish study performed by Orantes-Bermejo et al. also reported a non-correlation between presence of Echium pollen and PA/PANO levels in bulk honey samples [22]

-Also a recent Australian study performed by Carpinelli de Jesus et al. referred to children of 2-4 years having a honey intake of up to 28.6 g/day [29]

As far as the conclusions are concerned, we agree that they have been somewhat lost in the discussion. We have therefore restructured the last paragraph and added important conclusions. As a result, this paragraph now rounds off the discussion in a comprehensive manner. This paragraph now reads as follows:

As shown by the results of this study, honeys can endanger the consumer if PA/PANO-forming plants are predominant in the region and a virtually safe distance (> 3 km) to toxic plants did not guarantee safe honey. It has been shown that the PA levels of beehives at one site can vary greatly. Therefore, highly contaminated honey from a single bee colony could contaminate an entire batch. Considering the number of approximately 150,000 beekeepers in Germany [68], their families, and consumers of regionally produced honeys, it can be expected that more than 1 million consumers could face an elevated risk of PA/PANO exposure, because they face a worst case scenario due to high brand loyalty and higher intake. Additionally, it must be considered that the consumers’ interest in regionally produced and marketed products is increasing. This also applies to so-called ‘city honeys’, for which the botanical source of the honey can virtually not be controlled at all [29]. Consequently, it is highly recommendable to adequately respect regionally produced honey as a special source of PA/PANO and to identify beekeepers and consumers of regionally produced honey as a subpopulation with a higher exposure. Therefore, exposure and risk assessments should not only cover the average consumer of mixed retail honeys.

Reviewer 2 Report

I went over this carefully. I would probably make a number of minor editorial changes but in my opinion, this can be accepted as is.

This is a massive amount of work representing a well integrated set of studies.

Author Response

Thank you very much your time to review our manuscript for the positive feedback.

Reviewer 3 Report

Pyrrolizidine alkaloids are the most widespread toxins in natural origin. More than 6,000 plant species produce these toxic compounds. It’s also well-known that bees can forage on flowers of plants producing Pyrrolizidine alkaloids, which leads to contamination of honey with the toxic compounds.

Is Jacobaea vulgaris the major source of PA and PANO in honey products in Germany? Are there any other plants which could also contribute PA and PANO to honey products in Germany?

Besides honey, are there any other food products could cause the exposure of PA or PANO compound to consumer?   

This study used MOE method to calculate the health (or cancer) risk and follow the conclusion from EFSA CONTAM Panel in 2011. However, we also can find the TDI, PTDI or NOEL data in some earlier published papers. Did you bring into comparison between cancer risk assessment and non-cancer risk assessment for PA and PANO in honey products?

Although MOE of > 10,000 would be regarded as of little concern for an additional cancer risk of the consumer, some of scientific papers had discussed that reference value MOE of > 10,000 need to be reevaluated and adjusted case by case. What do you think about MOE threshold for Pyrrolizidine alkaloids?

Author Response

Thank you for reviewing our manuscript and your questions and remarks.

We will answer them point-by-point as follows:

-Is Jacobaea vulgaris the major source of PA and PANO in honey products in Germany? Are there any other plants which could also contribute PA and PANO to honey products in Germany?

RE: No, J. vulgaris is not the only PA source in Germany. PA/PANO of the Boraginacea type are found as well, originating from Borago, Echium, Symphytum, or Eupatorium species. These toxins, i.e. PA of the lycospamin type and echimidine were measured in the samples of this study and are presented in the results chapter. Their occurrence is also discussed in the discussion section. Therefore, we hope that these other toxins were sufficiently addressed. However, we agree that the introduction gives only little information on the occurrence of other species and therefore, the respective sentence was amended as follows: "In Middle Europe, indigenous Senecio species of the Asteraceae family such as S. vulgaris (common groundsel), S. aquaticus (marsh ragwort), the invasive neophyte S. inaequidens (narrow-leaved ragwort), and J. vulgaris represent important PA/PANO sources [46,47], along with Eupatorium spp. and certain species of the Boraginaceae family [26]" (5th paragraph of the introduction).

-Besides honey, are there any other food products could cause the exposure of PA or PANO compound to consumer?

RE: Yes, this fact is addressed in the introduction (2nd paragraph): In the light of food safety, (herbal) teas, culinary herbs and spices as well as food supplements can be contaminated with PA/PANO via co-harvesting parts or seeds of PA-containing plants and via horizontal transfer in the soil [12]. These food commodities are currently regarded as the main sources of human exposure to PA/PANO [4,13,14]. Food of animal origin like milk and meat may also contribute to the consumers’ exposure if animals are fed with PA/PANO-containing feed [4,15].

-This study used MOE method to calculate the health (or cancer) risk and follow the conclusion from EFSA CONTAM Panel in 2011. However, we also can find the TDI, PTDI or NOEL data in some earlier published papers. Did you bring into comparison between cancer risk assessment and non-cancer risk assessment for PA and PANO in honey products?

RE: This study refers to the current risk assessment approach of the EFSA (EFSA. Risks for human health related to the presence of pyrrolizidine alkaloids in honey, tea, herbal infusions and food supplements. EFSA Journal 2017) and of the German Federal Institute for risk assessment (Opinion No. 020/2018: Actual risk assessment concerning contents of 1,2-unsaturated pyrrolizidine alkaloids in food. BfR (Federal Institute for Risk Assessment): Berlin, 2018.). In the section 2.6. "risk assessment", both cancer (MOE-approach) and non-cancer effects (Health-based guidance value) are respected and evaluated. Results of both assessments are duly discussed and compared in the 7th paragraph of the discussion.

-Although MOE of > 10,000 would be regarded as of little concern for an additional cancer risk of the consumer, some of scientific papers had discussed that reference value MOE of > 10,000 need to be reevaluated and adjusted case by case. What do you think about MOE threshold for Pyrrolizidine alkaloids?

RE: Certainly, it is questionable in the case of carcinogenic substances such as PA, whether there can be a "safe" or "low risk" dose at all. In the manuscript it is therefore mentioned (1st paragraph of the introduction) that exposure to carcinogens should generally be as low as possible (EEC 315/93). However, the MOE evaluation is primarily intended to provide a recommendation for risk management, i.e. to show the need for action if the MOE value falls below a certain margin of exposure. This aspect was not discussed further, because I do not wish to interfere with the evaluation of the experts from EFSA and BfR in this matter.